# Differential Dominance of Ecological Processes Shapes the Longhorn Beetle Community in Tropical Rainforests and Temperate Forests of Southwest China

**DOI:** 10.3390/insects15030166

**Published:** 2024-02-29

**Authors:** Fang Luo, Tial C. Ling, Jacob D. Wickham, Farkhanda Bibi, Ana Gouveia

**Affiliations:** 1CAS Key Laboratory of Tropical Forest Ecology, Xishuangbanna Tropical Botanical Garden, Chinese Academy of Sciences, Mengla 666303, China; luofang@xtbg.ac.cn (F.L.); or tcling@ohio.edu (T.C.L.); 2Department of Environmental and Plant Biology, Ohio University, Athens, OH 45701-2979, USA; 3Severtsov Institute of Ecology and Evolution, Russian Academy of Sciences, 33 Leninsky Prospect, Moscow 119071, Russia; jacobwickham@gmail.com; 4Department of Botany, Abdul Wali Khan University Mardan, Garden Campus, Mardan 23200, Pakistan; farkhanda@awkum.edu.pk; 5Institute of Environment and Ecology, Tsinghua Shenzhen International Graduate School (SIGS), Tsinghua University, Shenzhen 518055, China

**Keywords:** Cerambycidae, community assembly, species coexistence, tropical and temperate forests

## Abstract

**Simple Summary:**

Our study investigated how different environmental factors influence the diversity and behaviour of wood-boring longhorn beetles (Cerambycidae) in Yunnan, China’s tropical rainforests and temperate forests. Yunnan is notable for its diverse plant species and varying landscapes. We conducted detailed surveys of beetle and plant populations across several forest plots, using various analytical methods, such as diversity comparisons and distance-decay relationships. We found a marked difference in beetle populations between the two forest types. The tropical rainforests were rich in beetle and tree species, with 212 Cerambycidae and 135 tree species identified. In contrast, the temperate forests had a much smaller population, with only 16 Cerambycidae and 18 tree species. This variation was mainly due to differences in environmental conditions and beetle movement limitations. The varied environment of temperate forests resulted in fewer beetle species and weaker interactions with plants, while the more consistent tropical rainforests supported a greater variety of beetles and stronger plant–beetle relationships. Our research highlights the significant impact of environmental factors, such as habitat variability and movement constraints, on the diversity and ecological interactions of Cerambycidae beetles. These findings are crucial for understanding and managing forest biodiversity, particularly in varying climatic zones.

**Abstract:**

(1) Background: Understanding the relationship between community assembly and species coexistence is key to understanding ecosystem diversity. Despite the importance of wood-boring longhorn beetles (Cerambycidae) in forests, factors affecting their population dynamics, species richness, and ecological interactions remain underexplored. (2) Methods: We surveyed cerambycid beetles and plants within five plots each across three transects in tropical rainforests and temperate forests of Yunnan, China, known for its rich biodiversity and varied elevation gradients. We explored a range of analytical tools, including *α*-diversity comparisons, distance-decay relationships, redundancy analysis, *β*-dissimilarity metrics, and various neutral community model analyses. (3) Results: The results revealed a stark contrast between the two forest types: the tropical rainforests hosted 212 Cerambycidae and 135 tree species, whereas the temperate forests had only 16 Cerambycidae and 18 tree species. This disparity was attributed to differences in environmental heterogeneity and dispersal limitations. In temperate forests, pronounced environmental variability leads to steeper distance-decay relationships and reduced α-diversity of Cerambycidae, implying stronger dispersal constraints and weaker plant–beetle associations. Conversely, the more homogenous tropical rainforests exhibited stochastic processes that enhanced Cerambycidae diversity and plant–beetle interactions. (4) Conclusions: Our findings underscore that environmental heterogeneity, dispersal limitations, and host-specificity are pivotal in shaping biodiversity patterns in Cerambycidae, with significant variations across climatic zones.

## 1. Introduction

Host-associated insect communities undergo shifts due to various deterministic and stochastic processes. Deterministic processes encompass ecological selection and involve the dynamic interplay of abiotic and biotic factors, representing distinct niches that influence community assembly [1,2,3]. Conversely, stochastic processes often result in niche overlap among species, demonstrating similar competitive capacities encompassing random birth, death, and ecological drift [4,5]. These dynamics critically shape species composition, causing the patterns that govern the assembly of species in a given forest to be visually indistinguishable from those emerging randomly [6,7]. Previous studies have shown that deterministic and stochastic processes concurrently regulate the assembly of ecological communities [8,9], although their relative importance could vary, depending on the environmental context [10]. This variation may arise from regulating deterministic and stochastic processes across temporal and spatial scales, which might be influenced by different intensities of ecological selection and shifts in the dispersal systems across various habitat contexts [11]. This underscores the need for an in-depth examination of community-driving forces across diverse habitats to enhance our understanding of the community assembly process.

Distance-decay relationships (hereafter referred to as DDRs) refer to the negative correlation between community similarity and escalating geographic distance. They provide a model for β-diversity variations across spatial scales, incorporating horizontal and vertical distances [12,13]. Both deterministic and stochastic processes are recognised for their influence on DDRs [6,14], and their ecological significance can vary across habitats [15]. Regarding the specific mechanisms, whether acting in combination or individually, biotic interactions, dispersal limitations, and ecological drifts, which relate to random fluctuations in species abundance within a community over time, have been found to shape DDR patterns [2,3,16]. For example, lower environmental heterogeneity, weaker biotic interaction, higher dispersal rates, and ecological drift tend to homogenise community composition, thereby diluting DDRs’ influence. Conversely, enhanced DDRs result from high environmental heterogeneity, intensified biological interaction, significant dispersal barriers, and diminished ecological drift [17].

The heterogeneity of environmental factors, such as air temperature, solar radiation, relative humidity, and exposure to wind, exhibits significant variations across different forests and climatic conditions [18,19]. Additionally, certain biological traits, including morphological features (e.g., body colour), phenological characteristics, and niche breadth (e.g., range of host-specificity), often vary across various habitats or regions [20,21,22]. As a result, the relative impact of niche- and neutral-based processes, through deterministic and stochastic factors, on community assembly and dynamics might fluctuate based on shifts in environmental conditions, spatial metrics, biotic interactions, and biological activities [23,24]. We can speculate that the ecological impacts of these mechanisms on the assembly of insect communities can vary based on host identities, topographies, spatial scales, and climatic zones [25,26]. These ecological impacts may have an even more pronounced effect on the assembly and composition of phytophagous insect communities that primarily feed on leaves, twigs, and various plant materials. Distinct DDRs are expected to emerge if environmental filters and biological interactions establish unique dispersal barrier ranges across heterogeneous habitats with diverse climatic conditions. This signifies changes in the importance of deterministic and stochastic processes in shaping beetle communities across varying forest types and climatic conditions. However, further investigation into these patterns is scarce.

Longhorn beetles (Cerambycidae), phytophagous insects of the Coleoptera order within the Insecta class, have cryptic larvae that feed on phloem and cambium during their early life stages (i.e., pine borers such as the *Monochamus* and *Arhopalus* genera) [27,28,29]. These develop within the xylem of their host plants, using it as a protective shield against adverse environmental conditions and predators [30]. They play important roles in forest regeneration and ecosystem cycling. With over 36,000 species from around 5300 genera, the Cerambycidae are among the most ecologically diverse and significant beetle groups globally [31]. Prior research has highlighted a considerable range of hosts, from healthy trees to coarse woody debris, exploited by the larvae of various beetle species [32,33]. This variability is especially notable across regions with distinct climatic types [34]. However, the significance of Cerambycidae’s diversity patterns and community assembly processes across these climate zones, such as those in the Yunnan province of China, remains to be clarified. Examining the contributions of deterministic and stochastic processes in the assembly of the cerambycid community could elucidate the ecological strategies of co-occurring species. Revealing the intrinsic linkages between assembly processes and species coexistence can facilitate the management of cerambycid communities for enhanced ecosystem service provisioning, such as forest regeneration.

This study investigated the changes in the composition and diversity of wood-boring longhorn beetle communities in the Xishuangbanna tropical rainforests and the Lijiang temperate forests of southwest China (Figure 1), which differed in terms of elevational gradients, plant composition, and environmental factors. In this study, we examined the Cerambycidae community assembly in forest ecosystems across different climate zones. In the subsequent sections, we discuss the implications of these findings, acknowledge the limitations of this study, and suggest potential future research directions. The present study aims to (I) evaluate the relative importance of deterministic and stochastic processes in shaping cerambycid communities across different climate zones and (II) to uncover associations between the assembly processes of cerambycid communities and the ecological strategies of co-occurring species across different climate zones.

## 2. Materials and Methods

### 2.1. Study Sites and Sampling Design

All field experiments were conducted in the Xishuangbanna tropical rainforests (hereafter referred to as XTRF) and the Lijiang temperate forests (hereafter referred to as LTF) in Yunnan province, southwest China (Figure 1). The XTRF is situated in southwest Yunnan (21.6° N, 101.5° E), at 600 m to 1000 m above sea level (a.s.l.). The region, characterised by a typical monsoonal climate, has clearly defined wet and dry seasons, from May to October and November to April, respectively. This area also exhibits an average annual temperature of approximately 22 °C, along with rainfall and precipitation measurements of around 1500 mm [35,36]. In contrast, the LTF is located in northwest Yunnan (27.1° N, 100.2° E), at altitudes ranging from 1015 m to 5596 m a.s.l. It experiences a mild climate, with the same distinct wet and dry seasons described above. This region has an average annual temperature of approximately 5.5 °C and an average annual precipitation of 1600 mm to 2500 mm [37,38]. The geographical distance between XTRF and LTF is approximately 300 km.

In each of XTRF and LTF, we established three transects with contrasting elevations, using the same interval (200 m elevation change) between each transect. In each transect, we laid out five plots, measuring 25 m × 20 m (Figure 1). Transects in the XTRF were established at 600 m (hereafter referred to as XTRF-T1), 800 m (XTRF-T2), and 1000 m (XTRF-T3) a.s.l. The transects in the LTF were set up at 3200 m (LTF-T1), 3400 m (LTF-T2), and 3600 m (LTF-T3) a.s.l. (Figure 1A,B). The distance between the transects was 1.3 km in the XTRF and 0.5 km in the LTF, with 50 m between the plots within each transect. Field experiments and sample collections were carried out from April 2018 to April 2019 in the XTRF and from May 2018 to May 2019 in the LTF. All the plots were positioned in areas unaffected by significant forest gaps resulting from anthropogenic or natural disturbances.

During our preliminary surveys (also see [34], we discovered longhorn beetles across different canopy strata. Thus, we installed window flight-intercept traps (hereafter referred to as FITs) in both the canopy and the understory to collect beetle samples. For constructing the FITs (cross vanes), we used two rigid transparent plastic plates measuring 50 × 35 cm (height × width) and approximately 5 mm thick. These plates were cut into rectangular shapes, following the design originally described by Ranius and Jansson [39] and further modified by Luo et al. [34]. The top-side plates were tied to transparent plastic plates (45 cm in diameter and 3 mm thick), and on the underside, they were attached to flexible plastic containers (35 × 30 cm, diameter × height; around 5 mm think). According to Shrestha et al. [40], beetles are more attracted to blue and yellow pan traps than colourless ones. Therefore, we used an equal number of blue and yellow containers for each plot. The FITs were tied to a branch approximately 1–1.5 m above the ground for the understory sample collection. To collect canopy samples, the FITs were positioned approximately 10–15 m above the ground using unmanned aerial vehicles and pulling ropes. Insect containers were filled with a 1:2 *v*/*v* mixture of 75% alcohol and an anti-freeze solution (ethylene glycol). Two traps were used for each plot; thus, 30 FITs were utilised for each forest type.

### 2.2. Data Collection

#### 2.2.1. Longhorn Beetle Sample Collection

Every ten days, we collected the longhorn beetles trapped in the collecting containers attached to the FITs. We replaced the collection basins and chemical solutions with new ones to prevent sample mixing. Once in the laboratory, all the collected beetle specimens were filtered using a portable transparent glass filter and preserved in 70% ethanol for taxonomic identification. Beetle specimens were identified to species level using taxonomic keys for Cerambycidae [41,42]. Voucher specimens were deposited at the National Zoological Museum of China, Institute of Zoology, the Chinese Academy of Sciences, Beijing, China.

#### 2.2.2. Plant Sample Collection

Between April and May 2019, we recorded and approximately identified the diameter at breast height (DBH) of each tree (with DBH ≥ 5 cm and height ≥ 1.5 m) in each of the five plots per transect in the XTRF and the LTF. We adhered to the guidelines set by the Center for Tropical Forest Science (CTFS; http://www.ctfs.si.edu/, accessed on 1 September 2019) and the Chinese Forest Biodiversity Monitoring Network (CForBio; http://www.cfbiodiv.org/, accessed on 1 September 2019) to assemble long-term, large-scale forest data from the tropics [43].

#### 2.2.3. Environmental Data Collection

We recorded seven environmental factors: annual mean temperature and humidity, annual temperature and humidity range, the maximum temperature of the warmest month, minimum temperature of the coldest month, and average elevation of each transect. Environmental data were recorded half-hourly using thermo-loggers (DS1923Hygrochron^®^ iButton^®^, Maxim, CA, USA) from April 2018 to May 2019. The thermo-loggers were attached to one of the FITs positioned in the canopy of each transect. Detailed information on the environmental data is provided in Appendix A.

### 2.3. Data Analyses

#### 2.3.1. Sampling Sufficiency

The sufficiency of our sampling from 30 samples (5 plots × 3 transects = 15 plots) per forest type within a 25 m × 20 m area, covering both beetle and plant communities, was assessed through coverage- and sample-based rarefaction curves [44]. The coverage-based rarefaction curve presents the proportion of total individuals in the community represented by sampled species, plotted against the collected number of individuals (or samples). Conversely, the sample-based rarefaction curve estimates the species count within a given number of observed individuals (or samples), thus reflecting sampling intensity. Extrapolation for both curves doubled the sample number, and 100-replicate bootstrapping was employed to establish a 95% confidence interval. We used the “iNEXT” package [44] to generate all rarefaction curves.

#### 2.3.2. Longhorn Beetle and Plant Community Composition and Environmental Variation

To assess differences in beetle community composition between the two forest types across different altitudinal gradients, we calculated various *α*-diversity indices, including species richness, Chao1 diversity, Shannon diversity, and Simpson diversity. We then applied a nonparametric Mann–Whitney U test to determine if significant differences existed. Additionally, to estimate the dissimilarity in beetle community composition between the XTRF and the LTF, we employed various β-diversity indices, such as the Horn index, Chao dissimilarity, and Sørensen distance. We computed a covariance matrix based on seven measured environmental factors, allowing for an assessment of their overall variations. The significant differences between the two forest types were confirmed using a Kolmogorov–Smirnov test.

#### 2.3.3. Influence of Plant Community, Environmental Factors, and Spatial Distance on Longhorn Beetle Community Variation

We performed variation-partitioning based on redundancy analysis to assess the proportional contributions (unique and shared) of each group of predictor variables (i.e., plant species composition, plant phylogeny, environmental metrics, and spatial distance) to the variation in beetle species composition. The significance of testable fractions (*p* < 0.05) was confirmed based on 999 permutations. To verify the differences in plant–beetle interactions across forest types under diverse climatic conditions, we followed the methods of Kemp et al. [45]. We computed the matrix of grouped plot-level *β*-diversity (Horn index) and partitioned it into various independent spatial components reflecting different *β*-diversity levels: (1) transects within plots (β_1_: 40–160 m scale), (2) transects between two neighbouring plots within a forest type (β_2_: 0.5–1.5 km scale), and (3) transects between two transects covering the highest elevational gradient within a forest type (β_3_: 1–3 km scale). The Mann–Whitney U test was used to assess the similarity of *β*-diversity for plants and beetles at each respective spatial scale (i.e., β_1_, β_2_, and β_3_). Additionally, we employed a linear regression model to examine the relationship between beetle *β*-diversity and plant phylogenetic *β*-diversity. For the plant phylogenetic *β*-diversity, we used the R package ‘plantlist’ [46] to obtain all enumerated species’ family and genus names. Further, we investigated their phylogenetic relationships using the online Phylomatic tool (www.phylodiversity.net/phylomatic/, accessed on 1 September 2019), following the Angiosperm consensus of Davies et al. [47]. Lastly, we constructed similarity matrices for the plant phylogenetic β-diversity, PhyloSor Index [48] using the ‘phylosor’ function in the picante package in R [49].

#### 2.3.4. Estimating Community Assembly Processes with Normalised and Modified Stochasticity Ratios

We utilised the Sloan neutral community model to examine the potential significance of neutral, stochastic processes in the beetle community assembly [50] by predicting the relationship between taxon frequency in local communities and their abundance in the broader metacommunity [9,51]. Within this model, we incorporated a parameter called “*m*” to represent the migration rate, which reflects the extent of dispersal limitation by estimating the likelihood of a random individual being replaced by dispersal from the entire community [52]. A higher *m*-value indicates lower dispersal limitation within the beetle community [53]. Furthermore, we calculated an R^2^ metric to evaluate the goodness-of-fit of the neutral model. The R codes used for model fitting and calculating goodness-of-fit statistics were adapted from previous studies [52,53].

We calculated the stochasticity (ST) ratio [54] and modified stochasticity (MST) ratio [55,56] to examine the underlying processes driving beetle community assembly (see also [31]). These ratios were computed using the ‘tNST’ function in the ‘NST’ package, with a threshold of 50%, used to differentiate between a more deterministic (below 50%) and a more stochastic (above 50%) assembly [57]. Considering the overall performance of similarity metrics, we utilised ST and MST based on Jaccard distance (STjac and MSTjac) to quantify the magnitude of stochasticity in the beetle community assembly [57].

To assess community assembly patterns, we employed a null model-based approach, the Raup–Crick dissimilarity metric, developed by Chase et al. [58] as an improvement on the original concept by Raup and Crick [59]. A Raup–Crick dissimilarity value close to 0 indicates a highly stochastic community assembly and significant community dispersal. Conversely, a value approaching −1 suggests that deterministic environmental filters shared across sites lead to similar communities [60]. On the other hand, a value closer to 1 implies that deterministic environmental filters favour dissimilar species compositions. The significance of differences in Raup–Crick dissimilarity between ST and MST was assessed using the Kolmogorov–Smirnov test (*p* < 0.05). Additionally, we constructed an ordination of community composition using nonmetric multidimensional scaling based on the Raup–Crick dissimilarity metric. Communities closer to each other exhibit greater deviation from null expectations, while more distant communities deviate less from null expectations.

#### 2.3.5. Software Program 

We performed all analyses and generated all graphs using R ver. 4.3.1 (www.r-project.org; R Core Team, [61]).

## 3. Results

### 3.1. Longhorn Beetle and Plant Surveys and Sampling Sufficiency Estimation

We recorded 1409 beetles belonging to 212 distinct species in the Xishuangbanna tropical rainforest (XTRF) and 1180 trees representing 135 species. In the Lijiang temperate forests (LTF), we observed 251 beetles from 16 species and 209 trees from 18 species. A comprehensive list of beetle and plant species observed in each forest type is shown in Appendix A (modified with permission from Luo et al. [34]).

Sample-based rarefaction curves indicated that the expected species richness was significantly higher in the XTRF compared to the LTF (Figure 2A,C). Furthermore, extrapolation analysis showed an increasing trend in species richness estimates for the XTRF, while the estimates for the LTF remained relatively consistent (Figure 2A,C). The coverage-based rarefaction curves indicated that our beetle and plant samples captured approximately 90% and 80% of the sample completeness for the XTRF and the LTF, respectively (Figure 2B,D).

### 3.2. Longhorn Beetle and Plant Community Composition and Environmental Variation Estimation

The beetle α-diversity, as measured by species richness, Chao1 diversity, and Shannon diversity, was significantly higher in the XTRF than in the LTF (*p* < 0.05; Figure 3A,D). The spatial variation in the beetle community structure, assessed using metrics such as the Horn index, Chao dissimilarity, and Sørensen distance, increased with the spatial extent (*p* < 0.05). Notably, these indices exhibited a steeper gradient in the LTF than the XTRF (Table 1). In terms of environmental variation, as determined by the covariance matrix of standardised environmental properties, the XTRF exhibited a significantly higher degree of variation than the LTF (*p* < 0.05; Figure 3E,F).

### 3.3. Influence of Plant Community, Environmental Factors, and Spatial Distance on Longhorn Beetle Community Variation

The redundancy analysis and variation partitioning showed that in the XTRF, the plant community composition, environmental variation, and spatial distance explained 11%, 2%, and less than 0% of the variation in the beetle community composition, respectively. These three factors explained 16% of the variation, while 77% remained unexplained (Figure 4A,B). In the LTF, these three factors individually explained about 4%, 7%, and 15% of the variation in the beetle community composition, respectively. When considered together, they accounted for 19% of the variation, leaving 57% unexplained (Figure 4A,B). Detailed information on the principal component (PC) axes of the most significant variables used to explain the beetle community composition can be found in Appendix A.

In the XTRF, the spatial distance influenced the composition of both plant and beetle communities. At the β_1_ scale, there was no significant difference in β-diversity between beetles and trees (H = 0.7102; *p* = 0.3993; Figure 4C,D). However, at both the β_2_ (H = 18.09; *p* = 2.11) and β_3_ (H = 36.06; *p* = 1.91) spatial scales, trees exhibited significantly higher β-diversity than beetles. Generally, the compositional β-diversity of trees and beetles followed similar patterns at shorter spatial distances but diverged as the distance increased, with this pattern being more pronounced in the XTRF.

The composition of both beetle and plant communities exhibited shifts at different spatial scales. The β-diversity value was significantly higher in trees compared to beetles across all β scales (H = 5.951; *p* = 0.0147 for β_1_; H = 55.74; *p* = 8.22 for β_2_; and H = 17.24; *p* = 1.91 for β_3_; Figure 4C,D). In the LTF, a clear trend emerged, where both beetle and tree species’ compositional β-diversity developed distinctively with increasing spatial scale, diverging more at larger distances. This pattern reflects a strong spatial structuring within the LTF, where the variety of beetle species is closely related to the diversity of tree species present. The statistical association between beetle β-diversity and plant phylogenetic β-diversity in LTF (estimate = 0.17, R^2^ = 0.147, *p* < 0.001; Figure 4E,F) suggests a more interconnected relationship between beetles and their host plants, likely due to the limited variety of available species in this temperate setting compared to the XTRF, where a similar yet stronger association was found (estimate = 0.22, R^2^ = 0.26, *p* < 0.001).

### 3.4. Estimating Community Assembly Processes with Normalised and Modified Stochasticity Ratios

The overall frequency of longhorn beetles exhibited significant variation in model fit across different macroclimatic habitats. The model fit was higher in the XTRF (R^2^ = 0.575) than in the LTF (R^2^ = 0.429; Figure 5). The estimated migration rate (m) followed a similar pattern, with the XTRF (m = 0.651) outperforming the LTF (m = 0.506). The proportion of the number of beetle species captured exhibiting neutral distribution was 97.2% (five species) in the XTRF and 87.5% (seventeen species) in the LTF (Figure 5A,B).

By applying the stochasticity ratio (ST) based on the Jaccard distance (STjac) index, we found that the beetle populations in both forest types were primarily governed by stochastic processes. The explanatory rates for this trend were higher in the XTRF (STjac: 86.98%) compared to the LTF (NSTjac: 71.39%; Figure 5C). Similarly, the modified stochasticity ratio (MST) based on the Jaccard distance (MSTjac) index showed a similar pattern, with higher explanatory rates in the XTRF (MSTjac: 75.94%) compared to the LTF (MSTjac: 56.56%) (Figure 5D).

These findings suggest that stochastic processes predominantly govern both climatic zones. However, the XTRF exhibited a greater increase in deterministic processes than the LTF. The median values of deterministic environmental filtering, based on the Raup-Crick dissimilarity index, differed significantly between the XTRF and the LTF (median value: −0.52 in XTRF and −1 in LTF; *p* < 0.05; Figure 5E). Furthermore, the nonmetric multidimensional scaling results revealed a higher level of similarity within the beetle communities in the LTF compared to the XTRF (Figure 5F).

## 4. Discussion

### 4.1. Coexistence Patterns of Cerambycidae across Different Climate Zones

In this study, we observed significant variations in the beetle community composition along elevational gradients, and these patterns were consistent across different forest types (Table 1). Notably, the slopes of DDRs were markedly steeper in the Lijiang temperate forests (LTF) than in the Xishuangbanna tropical rainforests (XTRF) of southwest China (Figure 1). Specifically, we identified deviations in five species within XTRF and seventeen species in LTF, a finding that enriches our understanding of the nuanced ecological dynamics influencing beetle diversity. These deviations not only reflect the significant role of environmental gradients and biological interactions but also underscore the importance of spatial factors in shaping the community composition of longhorn beetles. Changes in distance-decay relationships (DDRs) of similarity in plant and insect communities are often associated with dispersal limitations and deterministic processes, and they are influenced by various environmental and climatic characteristics [62,63]. These disparities can be attributed to differences in environmental heterogeneity (e.g., temperature and humidity) and biotic factors (e.g., insects and plants interaction) between the two regions [64,65]. The latitudinal biotic interaction hypothesis suggests that biotic interactions strongly influence communities in benign tropical environmental conditions (e.g., insulated from extreme temperature fluctuations with consistent seasonal variation). At the same time, abiotic characteristics become more prominent factors at higher latitudes, with extreme environmental conditions [66,67]. Environmental heterogeneity drives the species sorting and dispersal limitation, thereby intensifying DDRs [2,3,16]. In line with this mechanism, our results revealed a significantly higher spatial variation of standardised environmental properties in the LTF than in the XTRF. Our findings emphasised the impact of environmental heterogeneity on DDRs. Further, they provided insights into the differences in these relationships between forest types and elevational gradients, ultimately enhancing our understanding of biodiversity patterns and the factors driving them.

### 4.2. Effects of Biotic and Abiotic Factors on Cerambycidae

Habitats, particularly those experiencing simultaneous changes in biotic and abiotic factors, can significantly influence insects’ abundance and distribution patterns at both the species and community levels [68,69]. In our study, we found that the independent effects of environmental metrics and spatial distance on beetle community composition were prominent in the LTF compared with XTRF and contributed significantly to the observed variations in the beetle community composition in LTF. Our results suggest that higher environmental heterogeneity may intensify abiotic environmental sorting and limit the dispersal range of beetle community composition in temperate forests, intensifying the distance-decay relationship of beetle communities in temperate forests.

However, when employing various models, such as redundancy analysis (RDA), Horn-dissimilarity comparison across different spatial distances, and linear regression models, the beetle community in the XTRF exhibited a higher degree of host-specificity to plants compared to that in the LTF. In line with previous studies [45,70,71], we also observed a high level of association (symmetric Horn-dissimilarity distribution) over short spatial distances (i.e., β_1_; *p* = 0.3993) for both the beetle and plant communities in the XTRF. These results highlight plant communities’ important role in beetle community assembly over specific short distances in tropical regions. It also implies that longhorn beetles exhibit a higher degree of host-specificity in tropical areas than temperate ones. Still, their communities were less influenced by environmental metrics and dispersal limitations.

### 4.3. Effects of Deterministic and Stochastic Processes on Cerambycidae

Deterministic and stochastic processes are well documented in shaping plant and insect species composition and local assembly [72,73,74], reflecting their significant contributions to biodiversity maintenance and alteration by influencing longhorn beetle communities. Based on the neutral community model analyses, we discovered that the influence of neutral processes from tropical to temperate regions was notably decreased. Furthermore, the Sloan model has revealed that the migration rate of beetle communities was much higher in the XTRF than in the LTF. This higher migration rate in the XTRF, despite a marked host-specificity, suggests that beetles have developed specialized relationships with certain plants, facilitated by the forest’s biodiversity and intricate ecological networks. Such host-specificity does not preclude high migration rates but instead indicates beetles’ adaptive strategies to navigate the diverse and dynamic habitats of XTRF, to locate specific host plants. The heterogeneous distribution of these plants likely necessitates longer-distance migrations in search of suitable hosts. Conversely, the less complex ecological structure of LTF may not require or facilitate extensive migration among specialized beetles, reflecting a different set of ecological pressures that shape behaviour and adaptation strategies in temperate environments. This contrast underscores the nuanced ways in which ecological factors, such as biodiversity, plant distribution, and habitat complexity influence beetle migration patterns and host selection strategies across different climatic zones. Our findings contribute to a deeper understanding of cerambycid beetles’ ecological strategies, emphasizing the role of environmental heterogeneity and biological interactions in shaping their behaviours. The alignment of these observations with redundancy analysis (RDA) results corroborates our hypothesis, that tropical regions offer higher and less constrained dispersal ranges for Cerambycidae, in stark contrast to the significant dispersal limitations observed in temperate areas.

### 4.4. Different Coexistence Processes across Different Climatic Zones

Determining the linkages between community assembly and species coexistence is fundamental for understanding the mechanisms underpinning community diversity [9,75]. The contemporary coexistence theory emphasises that coexistence depends on niche and fitness differences [76]. In our study, the different distribution patterns of beetle community assemblages can be attributed to different coexistence processes within the two climatic zones. In the temperate zone, the ability of organisms to survive in changing environments often relies on their capacity to track shifting environmental conditions through migration or local adaptation [77]. Harsh highland climates, characterised by extreme cold, low oxygen levels, scarce precipitation, and intense ultraviolet radiation, will reduce the regional species pool and limit the migration capabilities of organisms [78]. Additionally, the divergence in species fitness is likely a primary driver for species coexistence. In our study, the presence of fewer species within the LTF, coupled with extreme environmental conditions, might lead to a reduction in ecological niches. However, this can result in an expanded species niche breadth for the existing species. In contrast, the presence of more benign and steadier environmental conditions in the study areas of the XTRF may offer greater resource availability and a larger species pool. These interactions, driven by niche processes, may promote convergence and divergence in key aspects of species’ ecological strategies [15], potentially narrowing species’ niche breadth. 

In addition, our study reveals that, as latitudinal gradients increase, deterministic processes gradually become a driving force to shape the structure of beetle communities, and the stochastic processes diminish. We can infer that the increased environmental heterogeneity in temperate forests is a driving force behind deterministic processes. In contrast, the stable climates of tropical rainforests would lend stochastic processes more power. Studies have shown that the stable climates of tropical rainforests support a higher diversity of species, facilitating the importance of biotic interactions, whereas the severe environment at higher elevations emphasises the importance of environmental filtering [79,80]. A few studies have verified the idea that stochastic processes may overwhelm deterministic processes in systems with less environmental variation [15]. Moreover, deterministic processes were less relevant to species co-occurrence association or interaction [52], coinciding with our result. In summary, this study underscored the significant role of environmental heterogeneity, dispersal limitations, and deterministic and stochastic processes in shaping the diversity and community assembly of wood-boring longhorn beetles in different climatic zones of southwest China, thus enriching our understanding of biodiversity dynamics.

### 4.5. Limitations of the Study and Recommendations for Future Research

Our research has limitations, as it was based on a study conducted in a narrow geographic region using a single taxon (the family Cerambycidae). Therefore, it is important to exercise caution when generalising these results to other contexts or ecosystems due to potential limitations. Future studies should aim to expand the geographic scope and number of site variables measured and encompass a broader range of taxa to validate the findings across diverse ecosystems and account for more of the variance. Furthermore, researchers should strive to integrate multiple dimensions of ecosystems, facilitating a comprehensive assessment of the respective influences of deterministic and stochastic processes on ecological communities. Adopting such an approach would significantly advance our understanding of the impacts of global change on biodiversity preservation and ecosystem dynamics in the Anthropocene.

## Figures and Tables

**Figure 1 insects-15-00166-f001:**
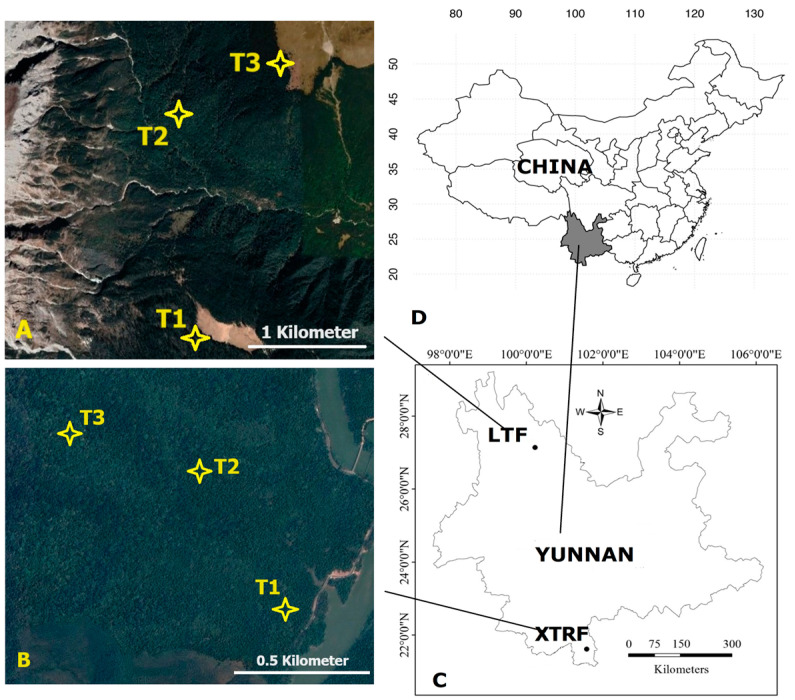
The geographical locations of 25 m × 20 m transects laid at (**A**) 3200 m (LTF-T1), 3400 m (LTF-T2), and 3600 m (LTF-T3) above sea level (a.s.l.) in the Lijiang temperate forests (LTF) and at (**B**) 600 m (XTRF-T1), 800 m (XTRF-T2), and 1000 m (XTRF-T3) a.s.l. in the Xishuangbanna tropical rainforests (XTRF) in Yunnan (**C**), southwest China (**D**).

**Figure 2 insects-15-00166-f002:**
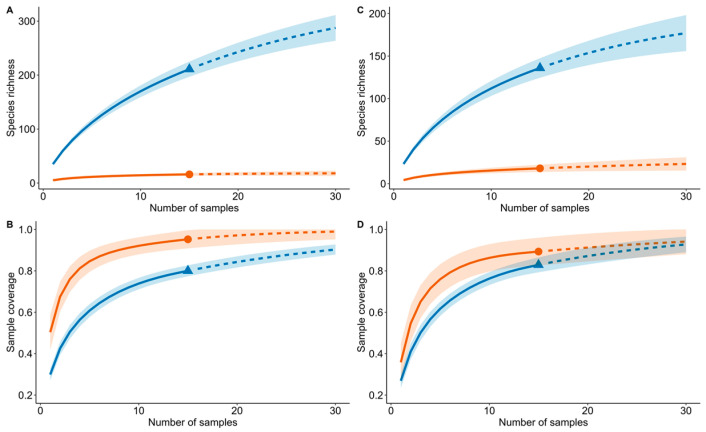
(**A**) Coverage-based and (**B**) sample-based rarefaction curves derived from beetle samples and (**C**) coverage-based and (**D**) sample-based rarefaction curves utilising plant samples. Sample collection was conducted in the Xishuangbanna tropical forests (orange colour) and the Lijiang tropical forests (blue colour) in Yunnan, southwest China. Solid lines represent the rarefied values, and dashed lines represent extrapolations up to a factor of two. Shaded areas represent the 95% confidence interval.

**Figure 3 insects-15-00166-f003:**
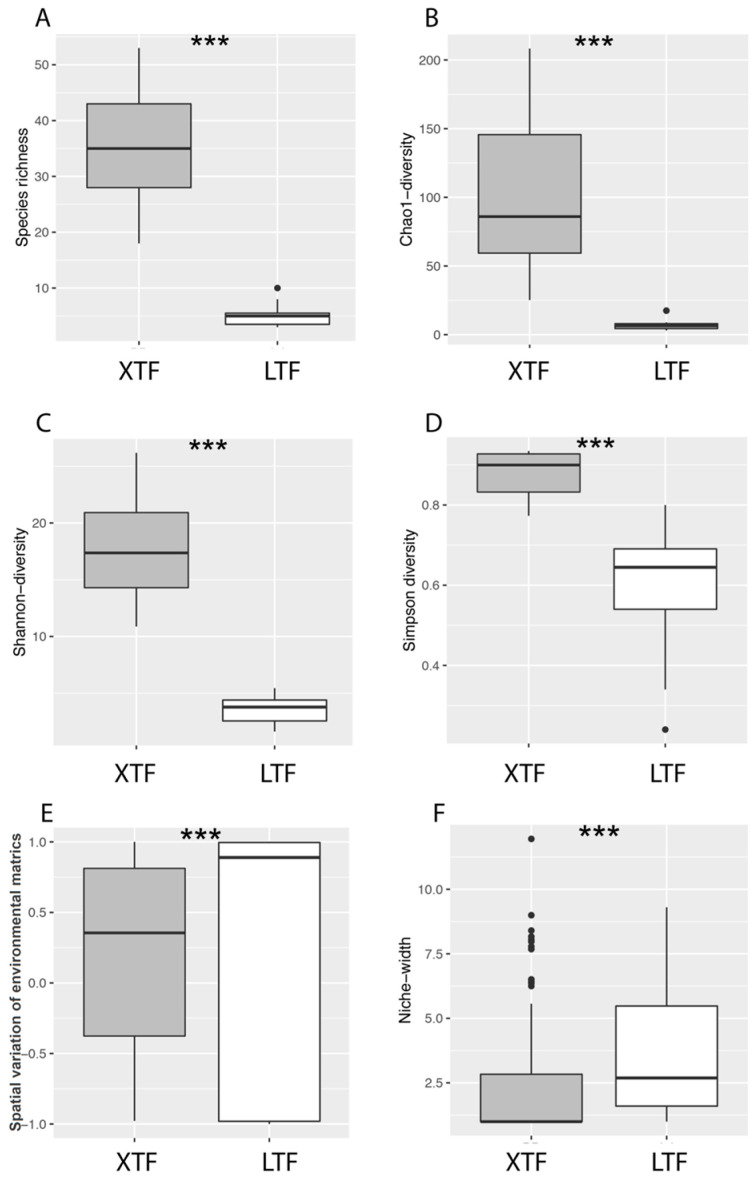
Comparative analyses of biodiversity and environmental variation metrics in the Xishuangbanna tropical forests (XTRF) and the Lijiang temperate forests (LTF). Figures (**A**–**F**) illustrate the following: (**A**) species richness, (**B**) Chao 1 diversity estimates, (**C**) Shannon diversity values, (**D**) Simpson diversity measurements, (**E**) spatial variations in environmental metrics, and (**F**) niche width estimations for both XTRF and LTF. The plots display medians as horizontal lines within the box, the interquartile range (i.e., 50% of all values) as the box limits, and 95% within brackets. The individual dots outside the upper whiskers represent outlier data points. Triple asterisks (***) denote significant differences between the forest types at *p* < 0.05.

**Figure 4 insects-15-00166-f004:**
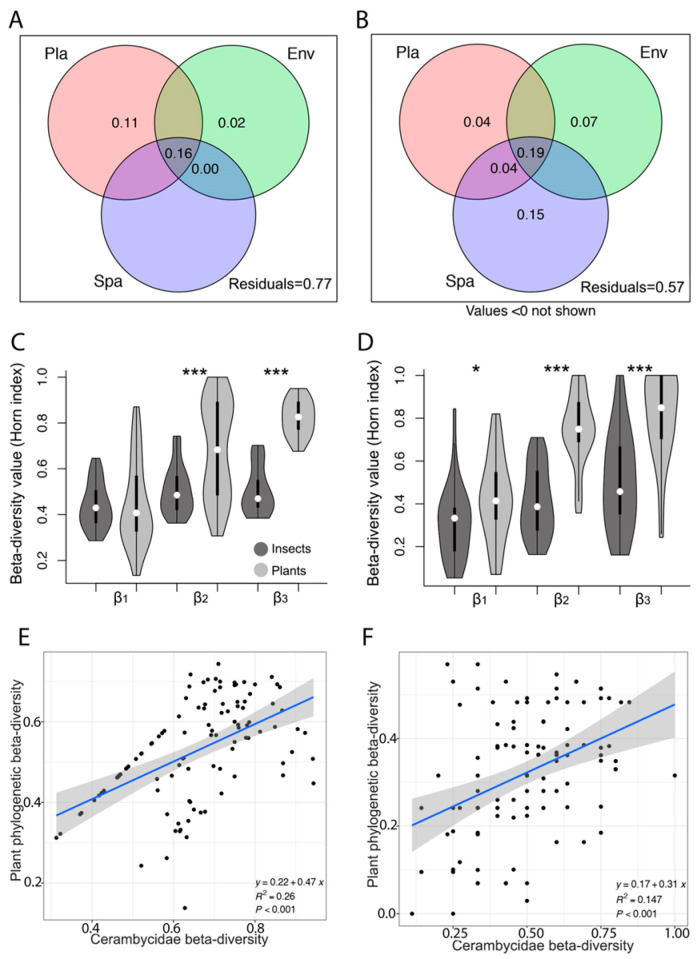
Analysing the determinants of longhorn beetle community composition and β-diversity in the Xishuangbanna tropical forests (XTRF) and the Lijiang temperate forests (LTF). (**A**,**B**) Variation partitioning analyses illustrate the independent and joint explanatory powers of plant species (Pla; pink circles), spatial distance (Spa; blue circles), and environmental variation (Env; green circles) on beetle community composition in the XTRF and the LTF. The percentage values within the circles represent the explanatory contribution of each factor. Negative values, if any, are not shown. (**C**,**D**) Presentation of *β*-diversity (1-Horn similarity) for both beetle (dark grey) and plant (light grey) species across transects (25 × 20 m) at different spatial scales in the XTRF and the LTF. The 1-Horn similarity index was calculated at three different scales: (i) within-transect plots (β_1_: 40–160 m scale), (ii) between neighbouring transects within a site (β_2_: 0.5–1.5 km scale), and (iii) between transects with the highest elevation difference within a site (β_3_: 1–3 km scale). Medians are denoted by white dots, first quartiles by thick black bars, and ranges by thin black lines. The plot shape corresponds to the data’s frequency distribution. Asterisks indicate significant *β*-diversity differences between beetles and plants at each spatial scale. One asterisk (*) denotes *p* < 0.05, while three asterisks (***) represent *p* < 0.0001. (**E**,**F**) Linear regression models depict the relationship between Cerambycidae *β*-diversity and plant phylogenetic *β*-diversity in the XTRF and the LTF.

**Figure 5 insects-15-00166-f005:**
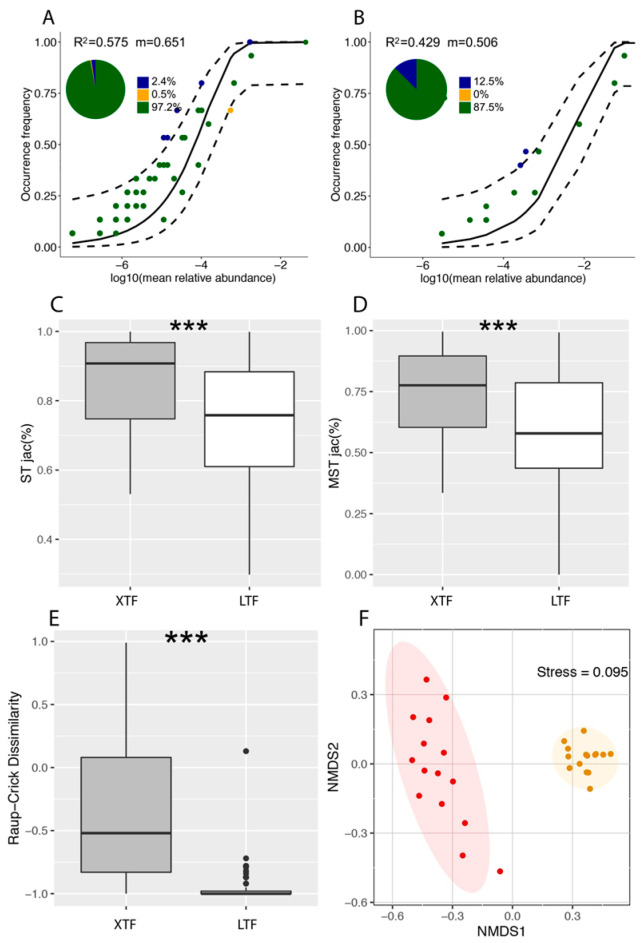
Comparative analysis of community assembly processes in the Xishuangbanna tropical forests (XTRF) and the Lijiang temperate forests (LTF). (**A**,**B**) These analyses use the neutral model for community assembly, with species occurrence frequency compared to model predictions. Species more prevalent than expected are marked in blue, while those less prevalent are denoted in yellow. Those meeting the model’s predictions are coloured green. Confidence intervals are displayed as dashed lines around the central prediction, represented by a black line. (**C**,**D**) Presentation of the normalised stochasticity test (NST) and the modified stochasticity test (MST), both developed based on Jaccard distance (NSTjac and MSTjac). A threshold of 50% distinguishes more deterministic (≤50%) from more stochastic (>50%) assembly in both forest types. (**E**) The Raup-Crick dissimilarity metric elucidates the balance between deterministic and stochastic processes in each forest type. A value of 0 signifies no variance from null expectations, whereas a value of 1 suggests observed dissimilarity exceeding null expectations, representing communities that are more different than expected by chance. Conversely, a value of −1 indicates communities that are less different than predicted by chance. (**F**) Nonmetric multidimensional scaling (NMDS) ordination of community compositions based on the Raup-Crick dissimilarity metric. Communities closer together deviate more from null expectations, while more distant communities deviate less. In this study, the XTRF communities (red symbols) deviate less from null model expectations, whereas the LTF communities (yellow symbols) deviate more significantly. Triple asterisks (***) denote significant differences between the forest types at *p* < 0.05.

**Table 1 insects-15-00166-t001:** Distance-decay relationships of beetle communities across elevational gradients in the Xishuangbanna tropical (XTRF) and the Lijiang temperate forests (LTF) in Yunnan, southwest China.

Sites	Measure	Horn Index	Chao Index	Sørenson Index
XTRF	slope	0.0003	0.0005	0.0003
	*p*	0.0116	0.0026	1.55 × 10^−8^
	R^2^:	0.0603	0.0847	0.26806
LTF	slope	0.0006	0.0006	0.0004
	*p*	0.0001	0.0023	2.84 × 10^−5^
	R^2^:	0.1336	0.0865	0.1571

## Data Availability

The data presented in this study are available on request from the corresponding author.

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
