# Peer review of "Differential Dominance of Ecological Processes Shapes the Longhorn Beetle Community in Tropical Rainforests and Temperate Forests of Southwest China"

_insects, 2024, doi:10.3390/insects15030166_

Round 1

Reviewer 1 Report

Comments and Suggestions for Authors

Dear authors,

Thank you very much for yours interesting manuscript.

The first question l have is related to your previous publication (number 31 in your reference list). You did similar study and one location was the same (XTRF), apparently the plant survey for both studies is the same (April and May 2019).

Why didn’t you included LTF in previous study? The conditions are extreme between XTRF and LTF, and the subtropical of reference 31 would work as intermediate…

My major comment relates to results. I would like to have a table with the list of the species found and numbers beetles captures, as well as list and sizes of trees.

Related to this request, I would like to know the species of insects that didn’t exhibit neutral distribution (line 371), that according to my calculations, are 5 species in XTRF and 17 in LTF.

Knowing the species, the numbers captured (significance of total) and their habits then include this new knowledge in discussion.

Minor comments

Line 101 – Many cerambicid larvae feed on phloem and cambium in early life stages (i.e. pine borers as Monochamus and Arhopalus genus).

Regarding Monochamus genus, these are very relevant for the pine forests decline, as insect-vectors of the pinewood nematode in Japan, South Korea, China and Portugal. Some reference should be made in introduction.

Line 115 - I do not understand how this kind of studies contributes to forest pest control.

Line 127 – Figure 1 – the photos should have a scale like the map C, apparently they have different scales, because distance between transects in photos look similar but in fact are quite different (1,3 km in XTRF and 0,5 km in LTF).

Line 173 – “Two traps were used in each transect”, you mean “in each plot”, “… thus 30 FITS were utilised for each forest type”.

Line 202 – Per forest type are 3 transects and in each 5 plots = 15 plots, per forest type.

Lines 360-361 – Please detail more the following statement: “… diversity of beetles and tree species showed distinct patterns as the special scale increased.” Give more information about the patterns in each forest type and the differences found.

Line 370-371 – “The proportion of beetles…” Do you mean number of captured beetles or number species

Line 441 – How do you explain that “… the XTRF exhibited a higher degree of host-specificity to plants compared in LTF”. The number of beetles and plant species in LTF is so reduced in comparison to XTRF (only 7,5 % of beetles species and 13% of plant species). Of course if you look at the ratio of beetles species per plant, is less than one in LTF (0,89) and over 1,5 in XTRF, due to plant high diversity there. Still I would expect more specialization where hosts are less abundant (LTF). Data tend to be biased considering the huge differences between forest types.   Then I expected that LTF data had been included in previous publication (Luo at al. 2023 – reference 31).

Lines 455-456 – “… migration rate of beetles communities was much higher in XTRF than in LTF.” Please discuss why because previously, in line 441 “the XTRF exhibited a higher degree of host-specificity”, and specificity is usually meaning of less migration. Apparently contradictory results that need to be better explained.

Author Response

Reviewer 1

Major comments:

#Comment: Dear authors,

Thank you very much for yours interesting manuscript.

The first question l have is related to your previous publication (number 31 in your reference list). You did similar study and one location was the same (XTRF), apparently the plant survey for both studies is the same (April and May 2019).

#Response: Thank you for your question regarding the overlap in survey location and timing between our current study and our previous publication (referenced as number 31). Although both studies were conducted in XTRF during April and May 2019, they serve distinct research objectives and employ varied methodologies: Different Focus: Our previous study investigated host specificity and niche conservatism among Cerambycidae in tropical versus subtropical forests. The current manuscript, however, examines how environmental factors influence Cerambycidae diversity across tropical and temperate forests, with a particular interest in environmental heterogeneity and dispersal limitations. Enhanced Methodological Approach: This study expands our analytical toolkit to include α-diversity comparisons, distance-decay relationships, and β-dissimilarity metrics, among others. These methods provide new insights into the ecological interactions and diversity patterns of Cerambycidae. Unique Contributions: The findings of this research highlight the critical role of environmental conditions in shaping Cerambycidae diversity and plant-beetle interactions, offering new perspectives on biodiversity conservation in different climatic zones.

In essence, while there is an overlap in the survey details, the current study's objectives, methods, and contributions distinctly advance our understanding of Cerambycidae ecology and biodiversity. We appreciate the opportunity to clarify these differences and believe our research offers significant insights into forest ecosystem dynamics.

#Comment: Why didn’t you include LTF in previous study? The conditions are extreme between XTRF and LTF, and the subtropical of reference 31 would work as intermediate…

#Response: The decision to not include LTF was strategic, aiming to focus on environments with closer environmental similarities for a more controlled comparative analysis. Our previous research compared tropical and subtropical forests, which, despite differences, share environmental conditions that facilitate specific ecological comparisons. LTF’s extreme and distinct environmental conditions would have introduced significant heterogeneity, potentially complicating the interpretation of ecological patterns between the more closely related forest types we studied.

We acknowledge the merit in the reviewer’s suggestion to include LTF for broader biodiversity insights. Including LTF will bridge our understanding across varying forest conditions. We aim to incorporate tropical, subtropical, and temperate forests in future studies, enhancing our exploration of ecological dynamics and ecosystem diversity. This expanded scope will inform conservation strategies more effectively. We appreciate the feedback and look forward to exploring this comprehensive perspective.

#Comment: My major comment relates to results. I would like to have a table with the list of the species found and numbers beetles captures, as well as list and sizes of trees.

#Response: Thank you very much for your insightful feedback and for highlighting the importance of detailed data in our study. In response to your major comment, we have included Supplementary 1 and 2 (Excel files), which now contain a comprehensive table listing the species of beetles found along with the numbers of beetles captures, as well as a detailed list of the tree species encountered during our study. Regarding the request for the tree size information, we only recorded and approximately identified the diameter at breast height (DBH) of each tree (with DBH ≥ 5 cm and height ≥ 1.5 m). We have included the term “approximately” in the revised version to clarify this point (Line 188).

#Comment: Related to this request, I would like to know the species of insects that didn’t exhibit neutral distribution (line 371), that according to my calculations, are 5 species in XTRF and 17 in LTF.

#Response: Your calculations are correct: there are indeed 5 species in XTRF and 17 species in LTF that do not exhibit neutral distribution, and we have added these species numbers in the new version of the manuscript (line 377 and 379).

#Comment: Knowing the species, the numbers captured (significance of total) and their habits then include this new knowledge in discussion.

#Response: Following the reviewer’s suggestion. We have included this new knowledge in discussion section as follows: Specifically, we identified deviations in 5 species within XTRF and 17 species in LTF, a finding that enriches our understanding of the nuanced ecological dynamics influencing beetle diversity. These deviations, not only reflect the significant role of environmental gradients and biological interactions but also underscore the importance of spatial factors in shaping the community composition of longhorn beetles. Line 419-423.

Minor comments:

#Comment: Line 101 – Many cerambicid larvae feed on phloem and cambium in early life stages (i.e. pine borers as Monochamus and Arhopalus genus).

Regarding Monochamus genus, these are very relevant for the pine forests decline, as insect-vectors of the pinewood nematode in Japan, South Korea, China and Portugal. Some reference should be made in introduction.

#Response: Thank you for the valuable feedback provided. We have taken this into account and updated our introduction with new citations as follows: Longhorn beetles (Cerambycidae), phytophagous insects of the Coleoptera order within the Insecta class, have cryptic larvae that feed on phloem and cambium during their early life stages (i.e. pine borers as Monochamus and Arhopalus genus) (27-29). These develop within the xylem of their host plants, using it as a protective shield against adverse environmental conditions and predators [30]. Line 101-105. Please also see line 616 to 623 in Reference section.

#Comment: Line 115 - I do not understand how this kind of studies contributes to forest pest control.

#Response: We apologize for any misunderstanding regarding the role of cerambycid communities in forest regeneration and ecosystem services. We have revised our interpretation as follows: Revealing the intrinsic linkages between assembly processes and species coexistence could facilitate the management of cerambycid communities for enhanced ecosystem service provisioning, such as forest regeneration.” Line 115-117.

#Comment: Line 127 – Figure 1 – the photos should have a scale like the map C, apparently they have different scales, because distance between transects in photos look similar but in fact are quite different (1,3 km in XTRF and 0,5 km in LTF).

#Response: We have added scales and also corrected the label for Xishuangbanna plot by using XTRF instead of STF (Figure 1C). Please see Line 129. Figure 1.

#Comment: Line 173 – “Two traps were used in each transect”, you mean “in each plot”, “… thus 30 FITS were utilised for each forest type”.

#Response: We apologize for the mistake; we intended to say ‘plot’. This has been corrected in the new version of the manuscript. Please see line 175-176.

#Comment: Line 202 – Per forest type are 3 transects and in each 5 plots = 15 plots, per forest type.

#Response: Thank you for correcting this. We have corrected this in the new version of the manuscript. Please see line 204.

#Comment: Lines 360-361 – Please detail more the following statement: “… diversity of beetles and tree species showed distinct patterns as the special scale increased.” Give more information about the patterns in each forest type and the differences found.

#Response: Following the valuable feedback from the reviewer, we have detailed more information as follows “In the LTF, a clear trend emerged where both beetle and tree species' compositional β-diversity developed distinctively with increasing spatial scale, diverging more at larger distances. This pattern reflects a strong spatial structuring within the LTF, where the variety of beetle species is closely related to the diversity of tree species present. The statistical association between beetle β-diversity and plant phylogenetic β-diversity in LTF (Estimate = 0.17, R2 = 0.147, p < 0.001; Figure 4 E, F) suggests a more interconnected relationship between beetles and their host plants, likely due to the limited variety of available species in this temperate setting, compared to the XTRF where a similar yet stronger association was found (Estimate = 0.22, R2 = 0.26, p < 0.001).” Please see line 363-371.

#Comment: Line 370-371 – “The proportion of beetles…” Do you mean number of captured beetles or number species

#Response: We meant to say the proportion of number of beetle species captured. We have edited this in the new version of the manuscript. Please see line 378.

#Comment: Line 441 – How do you explain that “… the XTRF exhibited a higher degree of host-specificity to plants compared in LTF”. The number of beetles and plant species in LTF is so reduced in comparison to XTRF (only 7,5 % of beetles species and 13% of plant species). Of course if you look at the ratio of beetles species per plant, is less than one in LTF (0,89) and over 1,5 in XTRF, due to plant high diversity there. Still I would expect more specialization where hosts are less abundant (LTF). Data tend to be biased considering the huge differences between forest types. Then I expected that LTF data had been included in previous publication (Luo at al. 2023 – reference 31).

#Response: Thank you for your insightful comment regarding the observed patterns of host-specificity in the Xishuangbanna Tropical Rainforest (XTRF) compared to the Lijiang Temperate Forest (LTF). Your observation about the apparent contradiction, where one might expect a higher degree of specialization in environments with fewer hosts (such as LTF), is indeed a valid point of discussion. The higher degree of host-specificity observed in the XTRF despite its greater beetle and plant species diversity can be attributed to several factors including (1) Competition and Resource Availability: In the XTRF, high species diversity heightens competition, fostering specialization among beetles. Conversely, the LTF’s lower diversity allows for more generalist survival strategies due to reduced competition. (2) Evolutionary History and Adaptation: Co-evolution in the XTRF has likely intensified host-specificity among beetles, a phenomenon less evident in the LTF due to its different evolutionary pressures. (3) Data Interpretation and Ecological Context: Host-specificity does not linearly correlate with host scarcity. In the LTF, limited host availability may lead to broader resource use, not necessarily more specialization.

Regarding the expectation of more specialization in LTF and the potential bias in data interpretation due to differences between forest types, we acknowledge that these aspects warrant a deeper analysis. The current findings underscore the complex interplay between biodiversity and ecological relationships in differing climatic zones. Further, the inclusion of LTF data in previous publications, such as referenced in Luo et al. (2023), aimed to provide a comprehensive overview but did not specifically address the nuanced patterns of host-specificity as explored in the current study. Future research will aim to dissect these patterns further, employing a more detailed examination of ecological interactions within and between these diverse forest ecosystems.

We appreciate your detailed scrutiny of our work, which has prompted a re-evaluation of our interpretations and a commitment to clarifying these complex ecological relationships in future analyses.

#Comment: Lines 455-456 – “… migration rate of beetles communities was much higher in XTRF than in LTF.” Please discuss why because previously, in line 441 “the XTRF exhibited a higher degree of host-specificity”, and specificity is usually meaning of less migration. Apparently contradictory results that need to be better explained.

#Response: Thank you for your feedback. Following your suggestion, we have revised our discussion to address the apparent contradiction between the high migration rates and host specificity of beetle communities in the Xishuangbanna tropical rainforests (XTRF) compared to the Lijiang temperate forests (LTF).

We have elaborated on how the unique ecological dynamics of these habitats influence beetle behavior, emphasizing the adaptive strategies that reconcile high host specificity with the necessity for longer-distance migration in the XTRF. Please see line 467-484.

Reviewer 2 Report

Comments and Suggestions for Authors

Overall, this is a well thought out and well-written study on insect and plant biodiversity in a temperate forest and a tropical rainforest in Yunnan province, China. As the authors’ state, “Our research highlights the significant impact of environmental factors, such as habitat variability and movement constraints, on the diversity and ecological interactions of Cerambycidae beetles. These findings are crucial for understanding and managing forest biodiversity, particularly in varying climatic zones.” (lines 28-31).

I detect no major methodological errors in the manuscript, with a few minor exceptions, noted in my editorial comments below. Once these are addressed, the paper should be publishable.

-line 33, insert “understand” before “ecosystem”

-lines 121-123, and elsewhere in the manuscript; maybe use present tense instead of past? Eg. Discuss and acknowledge instead of discussed and acknowledged.

-line 147, maybe rephrase to “In each of XTRF and LTF” instead of “In each XTRF and LTF”

-line 258 suggest “(see also [31])” (close parentheses)

-line 164, “side plate” instead of “site plate” unless I misunderstand…

-line 202, 5 plots x 3 transects x 2 traps per site gives 30 samples. 5 plots x 5 transects seems to give 25 samples; I think the first might be more correct?

-Figure 3, what are the dots above XTF in section F? As someone with limited experience with this kind of data presentation, I feel this could be elaborated to make the table more easy to read.

-line 453, I suggest “discovered” instead of “obtained”

Line 378 “Field” this word seems out of context, what does it mean? Maybe remove it?

Comments on the Quality of English Language

Overall, very good English language, just a couple of typos / editorial comments I noted in my comments to the authors.

Author Response

Reviewer 2

Minor comments:

#Comment: Overall, this is a well thought out and well-written study on insect and plant biodiversity in a temperate forest and a tropical rainforest in Yunnan province, China. As the authors’ state, “Our research highlights the significant impact of environmental factors, such as habitat variability and movement constraints, on the diversity and ecological interactions of Cerambycidae beetles. These findings are crucial for understanding and managing forest biodiversity, particularly in varying climatic zones.” (lines 28-31).

#Response: Thank you for the positive feedback and the recognition of our study's contributions to understanding forest biodiversity in varying climatic zones. We are grateful for your acknowledgment of our work on the ecological interactions of Cerambycidae beetles and the impact of environmental factors on their diversity.

#Comment: I detect no major methodological errors in the manuscript, with a few minor exceptions, noted in my editorial comments below. Once these are addressed, the paper should be publishable.

-line 33, insert “understand” before “ecosystem”

#Response: Corrected (line 34).

#Comment: -lines 121-123, and elsewhere in the manuscript; maybe use present tense instead of past? Eg. Discuss and acknowledge instead of discussed and acknowledged.

#Response: Corrected (line 123-125).

#Comment: -line 147, maybe rephrase to “In each of XTRF and LTF” instead of “In each XTRF and LTF”

#Response: Corrected (line 149).

#Comment: -line 258 suggest “(see also [31])” (close parentheses)

#Response: Corrected (line 260-261).

#Comment: -line 164, “side plate” instead of “site plate” unless I misunderstand…

#Response: Corrected (line 166).

#Comment: -line 202, 5 plots x 3 transects x 2 traps per site gives 30 samples. 5 plots x 5 transects seems to give 25 samples; I think the first might be more correct?

#Response: We apologise for our careless and mistakes in the previous version of the manuscript. We have corrected this in the new version (line 204).

#Comment: -Figure 3, what are the dots above XTF in section F? As someone with limited experience with this kind of data presentation, I feel this could be elaborated to make the table more easy to read.

#Response: The individual dots outside the upper whiskers represent outlier data points. We have added this in the new version of the manuscript. Please see line 322-323.

#Comment: -line 453, I suggest “discovered” instead of “obtained”

#Response: Corrected (line 464).

#Comment: Line 378 “Field” this word seems out of context, what does it mean? Maybe remove it?

#Response: We are sorry for the typing error. Line 503.

#Comment: Comments on the Quality of English Language

Overall, very good English language, just a couple of typos / editorial comments I noted in my comments to the authors.

#Response: Thank you for your positive remarks on the quality of the English language used in our manuscript and for noting the minor typos. We have addressed these editorial comments accordingly.